# Somatosensory Auras in Epilepsy: A Narrative Review of the Literature

**DOI:** 10.3390/medicines10080049

**Published:** 2023-08-21

**Authors:** Ana Leticia Fornari Caprara, Hossam Tharwat Ali, Ahmed Elrefaey, Sewar A. Elejla, Jamir Pitton Rissardo

**Affiliations:** 1Medicine Department, Federal University of Santa Maria, Santa Maria 97105-900, Brazil; jamirrissardo@gmail.com; 2Qena Faculty of Medicine, South Valley University, Qena 8362, Egypt; hossamtharwatali@gmail.com; 3Faculty of Medicine, Ain Shams University, Cairo 11835, Egypt; ahmed.h.elrefaey@gmail.com; 4Medicine Department, Alquds University, Jerusalem P850, Palestine; s.a.elejla@gmail.com

**Keywords:** aura, seizure, epilepsy, focal epilepsy, seizure semiology, EEG, MRI, somatosensory, paresthesia, painful

## Abstract

An aura is a subjective experience felt in the initial phase of a seizure. Studying auras is relevant as they can be warning signs for people with epilepsy. The incidence of aura tends to be underestimated due to misdiagnosis or underrecognition by patients unless it progresses to motor features. Also, auras are associated with seizure remission after epilepsy surgery and are an important prognostic factor, guiding the resection site and improving surgical outcomes. Somatosensory auras (SSAs) are characterized by abnormal sensations on one or more body parts that may spread to other parts following a somatotopic pattern. The occurrence of SSAs among individuals with epilepsy can range from 1.42% to 80%. The upper extremities are more commonly affected in SSAs, followed by the lower extremities and the face. The most common type of somatosensory aura is paresthetic, followed by painful and thermal auras. In the primary somatosensory auras, sensations occur more commonly contralaterally, while the secondary somatosensory auras can be ipsilateral or bilateral. Despite the high localizing features of somatosensory areas, cortical stimulation studies have shown overlapping sensations originating in the insula and the supplementary sensorimotor area.

## 1. Introduction

The term aura denotes the subjective symptoms at seizure onset that the patient perceives [1]. Aura means “breeze” and probably has its roots in Roman and Greek mythology, where “Aura” was a Goddess, a daughter of the Titan Lelantos and Periboea. She is raped by Dionysus and becomes the mother of twins. She maintains a blinding rage to the point of madness throughout pregnancy and tries to kill her children, which culminates in the murder of one of them. Then, a sequence of suicide attempts leads to her final plunge into the river Sangarius, where she is transformed into a spring [2].

In the 2nd century AD, Aretaeus (Aretaios), a physician born in Cappadocia, was one of the first to describe aura phenomena. His meticulous observational approach resembled distantly the modern medical scientific methods. He wrote books about several diseases, including epilepsy. He described “(…) an olfactory aura and sluggishness, vertigo, heaviness of the tendons, plethora and distension of the veins in the neck, and much nausea (Book 1, Chapter 5) and: But, if it be near the accession of the paroxysm, there are before the sight circular flashes of purple or black colours, or of all mixed together, so as to exhibit the appearance of the rainbow expanded in the heavens; noises in the ears; a heavy smell; they are passionate, and unreasonably peevish. They fall down then, some from any such cause as lowness of spirits, but others from gazing intently on a running stream, a rolling wheel, or a turning top. But sometimes the smell of heavy odours, such as of the gagate stone (jet), makes them fall down” [3,4]. One of the earliest descriptions of an aura has also been made by Galen, one of Aretaeus’ contemporaries, around 200 AD [5]. Historically, the aura was referred to as all prodromal symptoms before the seizure onset [6].

The International League Against Epilepsy (ILAE) characterizes an aura as a seizure with certain motor, sensory, autonomic, or psychic phenomena without losing consciousness [7]. It can be prolonged on rarer occasions, constituting a form of status epilepticus [8]. Aura is generally associated with focal aware seizures; however, it could occur right before focal impaired awareness seizures and in generalized seizures [9]. Auras can be experienced as negative or, more commonly, positive sensory symptoms. Positive symptoms include paresthesia, visual and auditory hallucinations, and somatosensory illusions. On the other hand, negative symptoms include numbness, deafness, blindness, and inability to move a body part [10].

The prevalence of aura can be considerably variable, ranging from 22% to 83% in epilepsy patients. Individuals with active epilepsy can report auras more frequently [5]. In a large study on 798 epilepsy patients, around 65% of patients reported having auras [11]. In Gower et al., a series of 2013 cases revealed that around 57% had an aura [12]. In another series of 1527 cases, about 56% of patients experienced an aura. The Epilepsy Phenome/Genome Project (EPGP) study revealed that women were more likely to report auras. However, this may be attributed to the larger number of older women compared to the other groups. Studying auras is extremely relevant as they can work as warning signs for people with epilepsy, increasing their sense of self-control of the disease and allowing them to take rescue treatments in seizure emergencies. Developing new drugs for rapid epileptic seizure termination is a pressing need, and the study of auras and prodromes in epilepsy plays an essential role in this field [11,13].

## 2. Methods

Six databases were used to find the studies in electronic form about somatosensory auras in epilepsy published from 1880 to April 2023. Science Direct, Excerpta Medica (Embase), Latin American & Caribbean Health Sciences Literature (Lilacs), Google Scholar, Medline, and Scientific Electronic Library Online (Scielo) were searched. The terms used for the search were “somatosensory, aura, epilepsy”. Publications in English, French, and Italian were included in the search. Our database search provided a total of 1671 papers; 1392 were irrelevant and 181 were unrelated to the review, duplicates, inaccessible electronically, or provided insufficient data. The detailed database search is provided in the Appendix A.

## 3. Auras in Focal and Generalized Epilepsy

Throughout history, the seizure type was usually identified based on the behaviors before, during, and after the seizure [6]. In this context, auras were used as indicators of seizures of focal origin that may progress to secondarily generalized seizures [14]. Focal seizures and auras were sometimes used synonymously in the 19th century [5]. Others considered an aura a type of focal seizure, whether it progressed to motor features or not [15].

Recently, the concept of the exclusive association of auras or prodromal symptoms with focal seizures has become controversial. The EPGP was a large study that revealed that around 65% percent of patients with generalized epilepsy reported the occurrence of an aura when asked about the symptoms using specific structured interview questions [11]. This is in accordance with previous studies that reported around 70% of patients with IGE (idiopathic generalized epilepsy) experienced auras. It was concluded that there was no difference regarding the frequency of aura symptoms between the focal epilepsy and IGE groups [14,15,16].

## 4. Underestimation of Auras

Noteworthy, only 21% of EPGP patients reported the occurrence of an aura when asked open-ended questions compared to 65% when asked structured questions [11]. This is relevant as physicians usually start the patient interview with open-ended questions and then may use specific closed questions to obtain further clinical details. Thus, history-taking is subjective and related to the clinician’s skills and background [11]. Moreover, closed questions could be less sensible for detecting auras as it has been wrongly determined that they are exclusive to focal seizures. Thus, on suspecting the generalized origin of seizures, the clinician may not ask specifically about the aura symptoms. This false belief may have led to underreporting of auras within generalized epilepsy. On the other hand, there is also a risk of overestimation using closed questions due to questions that are not being understood and false positive answers to please the healthcare professional. Additionally, answering open-ended questions depends on the patient’s knowledge and education level, in addition to the individual’s age, mood, mental status, and familiarity with the asked symptoms or condition [10,11].

In this way, the incidence of aura tends to be underestimated due to misdiagnosis by doctors or underrecognition by patients unless it progresses to an actual motor seizure. Additionally, some patients may not realize it is an epileptogenic event or prefer not to report it to the physician because they fear being judged or diagnosed with a psychiatric condition [17,18,19].

Somatosensory auras (SSAs) are a specific type of auras that are characterized by abnormal sensations on one or more body parts that may spread to other parts, which can resemble the somatotopic pattern “Jacksonian march” [19]. The most frequently reported somatosensory symptoms among epilepsy patients are paresthesia or tingling, followed by pain and thermal sensation [20,21].

The primary somatosensory area (SI), secondary sensory area (SII), and insula are the main cortical areas responsible for producing somatosensory perceptions (Figure 1). According to Penfield and Jasper, auras originating from the primary sensory area do not differ in character from those originating from the secondary sensory area. However, some patients in the latter group reported cold sensations [22]. However, Mazzola et al. revealed that paresthetic, pain, and thermal sensations could be elicited from all three areas [23]. Furthermore, upon stimulation of the insular cortex, non-somatosensory responses, such as auditory, gustatory, vegetative, etc., could be evoked [23,24].

The incidence of somatosensory auras is variable. Since the early patient series of Lennox and Gowers, they have been reported to be between 8–18% of patients with auras [10,25]. However, Mauguiere et al. reported a lower incidence of 1.4% in a study of one-hundred and twenty-seven patients [20]. Refer to Table 1 for the incidence of somatosensory auras throughout studies in the literature.

In a study with 798 individuals with GE, 530 reported bilateral tonic–clonic seizures. Of these, when asked open-ended questions, 112 (21.3%) reported auras. Regarding closed-ended questions, 341 participants (64.3%) experienced at least one form of aura [11]. Similarly, a cohort of 600 patients with focal epilepsy revealed that 12.5% of the patients had somatosensory auras [31]. Epilepsy with parietal foci had a higher incidence of SSAs, ranging from 25–73%. Frontal and temporal foci had a lower incidence of SSAs, up to 40% and 26%, respectively [21,31,32,33,34]. On the other hand, in occipital lobe epilepsy, no SSAs have been reported [35].

Silveira et al. assessed possible differences in seizure semiology among older and younger adults with epilepsy in the outpatient setting. The authors observed one individual which was more than fifty five years old and five individuals between 18–45 years old with SSAs. However, the results of the statistical analysis with SSAs were non-significant. Interestingly, subtle perceptions of transient confusion were seen in older patients but not in younger ones (*p* = 0.0028) [43]. These findings are important because they may contribute to greater difficulty in diagnosing seizures in the elderly.

The importance of studying auras also lies in their potential prognostication features. For example, in a prospective cohort study that included 498 patients from January 1999 to December 2017, the authors found that individuals with positive preoperative aura were 1.74-fold more likely to be seizure-free after surgery for temporal lobe epilepsy (TLE). Additionally, the data suggested a delayed onset of initial seizures in patients that presented with aura preoperatively (*p* = 0.04, 95% confidence interval (CI): 0.55–3.24). Additionally, individuals with less time since disease onset before TLE surgery were more likely to be classified as in remission after surgery. Preoperative sensory aura was a reliable predictor that a patient would achieve seizure freedom (*p* = 0.022) [44].

## 5. Localization Value of Somatosensory Auras

It was John Hughlings Jackson (1835–1911), an English neurologist, who first established a correlation between seizure symptoms and the location of the lesion [19]. Historically, before the era of advanced technologies and imaging modalities, surgeons primarily relied on ictal manifestation to identify the epileptogenic area for resection. Accordingly, auras, as the first symptoms of seizures, were an important localizing element of the epilepsy foci [45]. Later, intraoperative cortical stimulation provided a comprehensive method for epilepsy surgery [22,24,46]. Previous studies have established an association between somatosensory auras and the somatosensory homunculus. When there is an epileptic zone in the SI cortex, unilateral and distal sensory auras could occur. However, when the SII area is activated, this could result in bilateral symptoms [47].

There is an essential issue to remember in determining the location of lesions that cause seizures. Ictal discharges are generated in the epileptogenic zones in the brain and only manifest on spreading to other areas. Those areas which can produce symptoms are called symptomatogenic zones [48]. Thus, it is crucial to recognize the possible propagation pathways to be able to determine the area of origin of seizures. However, some symptoms may overshadow other minor clinical manifestations. Therefore, in patients who experience one type of symptom, it does not necessarily mean no ongoing ictal activities in other areas [10,49].

According to multiple studies, the most common localization for somatosensory auras is in the upper extremities, followed by the lower extremities and the face [20,31,50,51]. The hand specifically was the most frequent initial site, which reflects the wide cortical presentation of the hand [52]. 

In a French study by Mazzola et al., pain and somatosensory responses to direct intracerebral stimulations were acquired in subjects referred for epilepsy surgery. Percentages of somatosensory response in the SI (93.5%) and SII (83.0%) were significantly higher than that observed in the insula (64.0%) (*p* = 0.03) [23].

Palmini et al. investigated the localizing significance of auras in 179 patients with focal seizures. The study was performed with a retrospective and prospective cohort. In the retrospective group, 123 surgically operated patients had been followed up with electroencephalograms (EEGs) for four decades. The prospective group was comprised of 56 patients with DRE. The authors found thirty-two patients (twenty-four from the retrospective group and eight from the prospective group) with SSAs. One individual had temporal seizure onset, nine frontal, and twenty-two parieto-occipital from these. In the retrospective series, there was an association between SSAs and sites of seizure onset in one patient with temporal onset, eight patients with frontal, and fourteen patients with parieto-occipital onset. Seven out of eight patients with SSA in the prospective group had a final diagnosis of parieto-occipital epilepsy. Three individuals had only parieto-occipital spikes, two had parieto-occipital and temporal spikes, and the other had no interictal spikes [26]. 

Perven et al. retrospectively reviewed 333 patients submitted to TLE surgery between 2005 and 2010. From these, 26 (7.8%) individuals had SSAs. Ten patients with SSAs underwent invasive evaluation before surgery, with nine undergoing subdural grid with some depth electrodes. Only one patient underwent stereo-EEG. Almost half (twelve patients) had unilateral sensory symptoms, with three reporting symptoms ipsilateral to the epileptogenic zone, whereas nine patients reported contralateral symptoms. Among the most frequently reported sensations were tingling and numbness. Some individuals also mentioned “warmth”, “cold”, and “sizzling”. When compared to patients who were submitted to surgery straight away, patients who underwent invasive evaluation with subdural grids or stereo-EEG were more susceptible to presenting with seizures after surgery on multivariate analysis, although they sustained similar seizure freedom. According to the authors, these results may be due to poorer prognosis epilepsies in those individuals that require invasive EEG. According to the authors, these results may be attributed to more complicated epilepsies with discordant preoperative diagnostic data in patients who eventually required invasive EEG [27].

Yamamoto et al. reported a case of a 52-year-old right-handed Japanese man who had invasive EEG performed due to DRE presenting with focal seizures. The individual reported experiencing numbness in the upper left back that extended to the upper and lower left limbs. Brain magnetic resonance imaging (MRI) revealed a round calcified lesion in the right Sylvian fissure. Implanted subdural electrodes and ictal ECoG studies demonstrated low-voltage fast activities starting only from an electrode on the right inferior parietal lobule. There were no ictal activities observed in the parietal operculum electrodes. Somatosensory evoked potentials were detected in the ictal onset zone located caudal to the perisylvian area. Therefore, seizures originating from the inferior parietal lobe may result in focal seizures that exhibit ictal semiology and scalp EEG results similar to those originating from the second SSA. Noteworthy, this study showed that the sensory cortical representation has a high degree of connectivity with different brain areas [31,53]. 

A German retrospective study assessed localizing features of individuals with focal epilepsy who reported SSAs at seizure onset. Interestingly, all subjects reported somatic sensations described as dysesthesia, numbness, pain, temperature changes, and tingling. Of the 75 participants of the study with interictal and ictal surface EEG, 32% had parietal foci, 22% had a focus anterior from the frontocentral region, and 13% had temporal foci. In contrast, 20% did not localize the focus on the EEG well, 6% had vertex foci, and 5% had opercular foci. In 44 individuals, unilateral extremity auras were reported and were associated with a centro-parietal EEG focus in 36 patients and a lesion in that region in 30 patients. Bilateral extremity or axial auras were reported in 31 patients. From these, there was a correlation with a centro-parietal region EEG onset in twelve patients, temporal cortex in six, vertex in seven, and diffuse foci in six patients. In 38 individuals, the SSA remained localized without march, followed by a motor pattern. In 30% of the patients, postural tonic and psychomotor seizures occurred. In only six percent of the individuals, unilateral tonic seizures were associated with the SSA in the upper extremity and a contralateral central parietal focus. Subjects with diffuse or hemispheric lesions reported negative motor features. In 17%, generalization with tonic–clonic motor features was observed. In thirty-seven patients, the SSA progressed with a march, occurring in the upper extremity in 46% and leg in 12%. A hemicorporal spread occurred in 24%, and a truncal spread in 14% of the subjects; however, contralateral spread over the midline was rare and was only seen in 2% [31].

Matsumoto et al. reported an adult man with focal cortical dysplasia (FCD). They explored epileptogenesis mechanisms by paired-pulse direct cortical electrical stimulation. Electrocorticograms (ECoGs) were performed, and corticocortical evoked potentials (CCEPs) were obtained by averaging ECoGs recorded from the surrounding areas. Subdural electrodes were placed for invasive monitoring of the patient’s left foot primary somatosensory (SI) and motor areas. The patient experienced an SSA, which evolved into a left leg clonic seizure. During this period, single and paired stimulation at the focus showed increased cortical excitability defined by enlarged CCEP and decreased intracortical inhibition, which suggested increased intrinsic epileptogenicity during seizure generation [54].

A Korean study investigated the clinical features and the diagnostic sensitivity of brain MRI, interictal/ictal scalp EEG, positron emission tomography with 18-F-fluorodeoxyglucose (FDG-PET), and ictal single-photon emission computed tomography (SPECT) in patients with parietal lobe epilepsy (PLE). Forty patients who were diagnosed with PLE between 1994 to 2001 were included. To diagnose PLE, there should be either a discrete lesion in the parietal lobe on brain MRI with ictal EEG or an exclusive ictal onset zone in the parietal lobe confirmed by intracranial EEG. A total of 27 patients underwent surgery. Surgical outcomes were defined as “seizure-free” or “non–seizure free” and “favorable” or “unfavorable”. Seizure-free was considered when there was seizure remission after surgery, including the absence of auras. The postoperative seizure frequency should decline by more than ninety percent to be considered a positive outcome. A video EEG monitoring system recorded interictal/ictal scalp EEGs in all patients. Intracranial EEG monitoring was performed in twenty-seven cases with inconclusive or discordant results. The placement of the grid or strip was determined by the results of ictal scalp EEG, PET, ictal SPECT, and clinical semiology. Pre- and intraoperative functional mapping and intraoperative ECoG were also performed when necessary. A total of 27 of the 40 patients experienced at least one type of aura before a seizure [28]. 

The Korean study showed that SSAs were the most frequently (13, 32.5%) reported auras. SSAs were contralateral to the side of the seizure in ten (25%), bilateral in two (5%), and ipsilateral to the side of the seizure in one (2.5%) patient. The affective aura was the second most frequent type, which was experienced by six (15%) patients, followed by vertiginous (four, 10%), visual (four, 10%), autonomic (three, 7.5%), and gustatory (three, 7.5%) auras. Among the forty patients, twenty-seven underwent operative surgery. Parietal neocortical resection was done in twenty-one subjects, simple lesionectomy in five patients, and lesionectomy with marginectomy in one patient. Among the twenty-six patients who underwent surgery with more than one year of follow-up, twenty-two (84.6%) had a favorable surgical outcome, including fourteen (53.9%) seizure-free patients, whereas four (15.4%) patients had an unfavorable outcome. Ictal EEG correctly localized the lesion in five of the fourteen seizure-free patients and five of the twelve non-seizure-free patients. Ictal EEG lateralized the epileptogenic hemisphere in six patients. Ictal EEG was falsely localized to the temporal, occipital, or frontal electrodes in seven individuals. Non-lateralization was found in one patient [28]. In 14 remittent patients, interictal scalp EEG did not localize the epileptogenic lobe. In four patients, interictal EEG lateralized the epileptogenic hemisphere. However, seven individuals presented with non-lateralization, including three patients with a normal interictal EEG. Additionally, interictal EEG was falsely localized to three patients’ temporal or frontal electrodes [28]. 

Salanova et al. assessed 82 patients with non-tumoral PLE treated surgically at the Montreal Neurological Institute between 1929 and 1988. The study’s main objective was to identify the clinical manifestations, ECoG, results of cortical stimulations, and prognostic factors. Pre-excision ECoG was performed in 63 patients and postresection in 46 patients. A total of 80 patients underwent intraoperative cortical stimulation. ECoG was continued during stimulation. It was found that 94% exhibited aura, of which SSAs (55.3%) were the most frequently reported. The epileptogenic zone was contralateral in fifty-one individuals and bilateral in one individual. Thirteen of the fifty-two subjects described pain, and five reported a thermal sensation. Nine subjects reported disturbances of body image, of which 33.3% mentioned a sensation of movement in one extremity, and one individual described the absence of one leg. A total of 66 patients had surface EEG, and seizures were recorded in 36 [45]. 

The fronto-centro-parietal region (22 out of 66, 33%) was the most commonly observed location of interictal epileptiform discharge. The other areas associated with abnormal electrical activity were the parietal-posterior-temporal (14%), parietal (14%), parieto-occipital (9%), fronto-centro-temporal (4.5%), fronto-temporal-parietal (4.5%), hemispheric maximum posterior head region (9%), bilateral (4.5%), and no epileptiform discharges (7.5%). On the other hand, secondary bilateral synchrony was described in 32% of the individuals. Most cases revealed a lateralized ictal discharge, but it was also observed over the centro-parietal and posterior head regions. In four patients, there was localized parietal seizure onset. A total of 80 patients underwent intraoperative cortical stimulation. Auras were reproduced in 44 out of 80 (55%) patients, which were more commonly referred to as a tingling sensation. Furthermore, 43 patients had right parietal corticotomies. Pre-operative ECoG showed spiking in the following regions: superior parietal (eleven), inferior parietal (eleven), fronto-centro-parietal (five), and opercular region (nine), no epileptiform discharges (two), and unavailable data (five). In seven out of thirty-six (19%) patients, spiking was also recorded from the posterior-superior-temporal region [29].

Left parietal corticotomies were performed in 39 individuals, but only 25 had ECoG data. The spiking patterns were: superior parietal (14), inferior parietal (one), opercular region (four), upper supramarginal gyrus (two), posterior-temporal-parietal (one), posterior parietal (one), parieto-occipital (one), and no epileptiform discharge (one). Post-resection ECoG was available in 46 patients: 26 (57%) had either no spiking or a significant reduction of spiking, and 20 out of 46 (43%) had residual spiking. Furthermore, 79 individuals had a complete description of the follow-up, in which 23% became seizure-free after early attacks, 21.5%% had worthwhile improvement, 20% were seizure-free since surgery, 14% had no improvement, 10% had rare seizures since surgery, 9% were initially seizure-free with late recurrence of rare seizures, and 2.5% had only auras. Therefore, 65% of patients benefited significantly from surgery [29].

Afif et al. studied intracerebral electrical stimulation in patients with epilepsy to evaluate the anatomo-functional organization of the insular cortex. Twenty-five individuals with drug-resistant focal epilepsy were submitted to insular cortex stereotactical implantation of at least one electrode. Sixty-seven stimulations induced at least one clinical response. Eighty-three responses were evoked from insular cortex stimulation. The main responses were sensory, motor, pain, auditory, oropharyngeal, speech disturbances, and neurovegetative phenomena. Regarding somatosensory responses, 10 patients mentioned 11 responses (13.2% of all responses and 28.2% of responses evoked by stimulation sites within the posterior insula) contralateral to the electrical stimulation (ES) site. Three responses were reported in the upper limbs, three in the lower limbs, and one in the neck. Five of these seven responses were evoked by ES in the non-dominant hemisphere, the remaining two in the dominant hemisphere. Electrodes located in the postcentral insular gyrus were responsible for all responses. Following stimulation, four warmth sensations were reported by four patients. From these responses, three were achieved by ES of the postcentral gyrus, two occurred in the cranium, and one in the contralateral elbow. The ES of the middle posterior long gyrus of the posterior insula resulted in one response where the individual mentioned warmth in all four limbs [30].

## 6. Subtypes of Somatosensory Auras

### 6.1. Paresthetic Auras

Paresthetic aura is the commonest subtype of somatosensory auras, which can present as abnormal sensations in the body, like tingling, numbness, or a feeling of electrical shocks [36,55]. The most common sensory alteration experienced in PLE is paresthesia. It is usually considered a premonitory symptom that can occur before a seizure or migraine headache. Sensory aura was reported to be the most common type of aura. Additionally, the paresthetic aura is the most common type of sensory aura that can be present in body parts, although it is most experienced in the arms, legs, and face [55].

Studies on individuals with epilepsy have found that paresthetic aura is commonly reported, with a prevalence of approximately 20% in individuals with epilepsy [18,55]. An extensive case series on 1359 epilepsy patients revealed that paresthetic symptoms were reported in about 7% of all patients with an aura [10]. A study of 127 patients has reported that more than 89% of patients with SSAs had experienced paresthesia. Another study reported that tingling or numbness was the commonest sensory symptom, with over 82% [21,31]. It was also found that paresthetic aura is more commonly experienced by women and often occurs in people with a family history of epilepsy [36,56].

The most common manifestation of paresthetic aura was tingling (89%), followed by sensory loss or numbness (11%) [56]. However, it can be described in several other ways, such as tickling, prickling, a feeling of pins and needles, the sensation of electric shock, or something below the skin [22]. 

The exact cause of paresthetic auras is not fully defined, but it can be related to abnormal activity in the brain in conditions such as epilepsy and migraines [56]. It can occur as an isolated symptom or in combination with other aura symptoms. A study by Reed et al. highlighted that paresthetic auras are often accompanied by other aura symptoms, including visual or auditory disturbances, but they occur mainly in cases with migraines. The authors mentioned that the duration of paresthetic auras could vary significantly, from only a few minutes up to an hour [57]. Similarly, a paresthetic aura can stimulate the cortex to evoke another type, such as a painful aura; emitted discharges of escalating intensity can propagate to elicit a different type of sensation, which forewarns an impending seizure [58].

In previous studies, paresthetic sensations were elicited mainly after stimulation of the anterior part of the postcentral opercular cortex and were distributed in the SI and insula. After SI stimulation, evoked responses in limbs were predominantly contralateral. However, after stimulation of SII and the insula, they could be ipsilateral or bilateral. The sensations evoked by stimulation of the face or trunk were mostly bilateral, regardless of the region stimulated. The response lateralization was similar for both left and right stimulation [47].

In this context, paresthetic auras have many clinical implications. In a stereo-EEG study, direct electrical stimulation of the insular cortex was performed in patients with TLE candidates for neurosurgery. Interestingly, a sequence of ictal symptoms was found to be reliable enough to characterize insular lobe epileptic seizures, which included laryngeal discomfort with thoracic oppression or dyspnea, followed by unpleasant paresthesia or a warmth sensation focused on the perioral region or extended to a large somatic territory and dysarthric or dysphonic speech, reflecting a propagation of the discharge to the insular cortex. According to the authors, this finding has important practical applicability regarding surgery outcomes, especially if this ictal sequence occurs at the beginning of the stereo-EEG discharge or later in the seizure. In this context, insular resection of the insular focus could lead to seizure remission in individuals with seizures from the insula. On the other hand, a late presentation of this ictal sequence in people with TLE can suggest that the conventional anterior temporal lobectomy is more effective [59].

### 6.2. Painful Auras and Migraine

First reported by Gowers in 1901 [12] and again thirty years later by Lennox and Cob [10], painful epileptic auras can be generally classified according to the anatomic location of the sensation felt, whether it is peripherally localized (painful somatosensory seizures), cephalic, or abdominal [60]. In painful somatosensory seizures (PSS), patients usually report an acute and intense pain (burning sensation, pricking ache, throbbing pain, or muscle tearing sensation) affecting part of the body, which can be reported lateralized, contralateral, or ipsilateral to the epileptic focus. Despite the early reports of painful auras, their prevalence is usually low, affecting less than one percent of the individuals with DRE [61]. In a study of focal epilepsies, only 3% of patients reported painful auras [31]. ECoG studies of painful somatosensory auras localize the epileptogenic zone to the posterior insula and secondary somatosensory cortex (Figure 2) [30,47,62].

Despite extensive investigations, the differential diagnosis of pain rarely ever includes epilepsy. However, oversight of an ictal origin of the pain can delay treatment in many patients, leading to misdiagnoses and poor outcomes. Accordingly, patients reporting paroxysmal abdominal pain, which is subjective to many clinicians, must be observed for disturbance in the state of consciousness, assessed for epileptiform EEG abnormalities, and evaluated for a positive response to antiepileptic therapy [63]. Nonetheless, investigating painful auras holds paramount localization and lateralization value in the early management of subjects with epilepsy, which could influence diagnostic testing, management plans, and prognostic characterization.

Moreover, according to Nair et al., abdominal pain was reported in 5% of abdominal auras seen in TLE and 50% of abdominal auras seen in frontal lobe epilepsy. The most common sites of localization were found to be the peri-umbilical or upper abdominal areas, both of which are represented profoundly in the nervous system. Seizures with parietal lobe origin frequently manifest with pain, which is often a reliable indicator of their presence [64].

In a cortical stimulation study, the onset of painful somatosensory seizures was seen systematically after stimulation of the insula and SII area, and in a lesser frequency, after stimulation of the opercular portion of the SI area. Interestingly, ictal pain was consistently reproduced by stimulating the insula or SII region but never by stimulating the SI area. The authors emphasized that paroxysmal lateralized pain can be the only manifestation of epileptic seizures arising from the operculum-insular cortex, and it could be misdiagnosed as psychogenic manifestations or radicular pain. Thus, knowing this singular form of sensory seizures is important for appropriate clinical management [61].

In addition to the theory of electrical stimulation, others were more specific (Figure 3). For example, contralateral involvement of the primary somatosensory cortex is thought to induce unilateral peripherally localized painful auras [58].

As for the source of painful auras involving the abdomen, the amygdala was found to communicate with the gastrointestinal system through the dorsal motor nucleus of the vagus nerve [65]. Other studies have speculated on the role of the insula and Sylvian fissure in the development of viscerosensory auras. However, conclusive evidence of the etiology of painful auras is yet to be explored [66].

On the other hand, painful cephalic auras [63] are postulated to have their origin in the secondary somatosensory cortex [67]. It is important to distinguish painful cephalic somatosensory auras from ictal epileptic headaches. While painful cephalic auras are caused by an electrophysiologic disturbance involving a localized cortical somatosensory area responsible for pain, ictal headaches are thought to be caused by a sub-clinical epileptic discharge activating the trigeminovascular system, which can cause migraine/headache without any other associated cortical epileptic sign or symptom [68]. Ictal headaches also do not have localizing value in most cases, but in patients with TLE, they could be ipsilateral to the seizure onset. Some articles have proposed diagnostic criteria for ictal epileptic headaches: (a) headache lasting minutes, hours, or days. Headache that is ipsilateral or contralateral (if the headache is not generalized) with epileptiform EEG discharges; (b) variable EEG abnormalities may be observed without a specific EEG or clinical headache pattern required; (c) headache and EEG abnormalities immediately resolved after antiepileptic drugs intravenous administration [1,63,69,70,71,72].

Gowers remarkably noticed in the 20th century that “migraine is in the borderland of epilepsy”, suggesting common or overlapping underlying pathophysiological mechanisms between these conditions [73]. In 1858, Edward Henry Sieveking published an essay, “On Epilepsy and Epileptiform Seizures,” where he defined “cephalagia epileptica” as “(…) indicative at once of what I regard as a cause and a tendency, occurring possibly in a subject in whom the epileptic paroxysm has been manifested merely by slight vertiginous attacks, by a single attack in former times, or by some spasmodic action that alone would not be regarded as an epileptiform character. The pain in this form of headache may affect any part of the head, but it is frequently limited to a spot at the vertex; and where that is the case, I have found marked benefit arise from making the attack directly upon the apparent seat of injury [74,75]”. 

According to a historical study by Mervyn J. Eadie [76], when Sieveking published his study, the headache was only known as an isolated occurrence, not associated with auras, regarded as a main characteristic of seizures. Thus, Sieveking probably described a series of cases of patients with migraine aura evolving to seizure, which would be years later defined as “migralepsy [76,77]”. 

The term “migralepsy” was created by Dr. Douglas Davidson, and in 1960, Lennox et al. [78,79] reported a case of “ophthalmic migraine with perhaps nausea and vomiting followed by symptoms characteristic of epilepsy”. Even though the name suggests a simultaneous occurrence of migraine and epilepsy, it describes a clinical manifestation where a migraine aura evolves into an epileptic seizure within an hour [78]. The term became popular among researchers in the 1960s, but it also received criticism, mainly in the last two decades, for leading to possible confusion, misclassification, and misdiagnosis. Despite the controversy, the 3rd edition of the International Classification of Headache Disorders recognizes this clinical entity under “migraine aura-triggered seizure”. It is described as “a seizure triggered by an attack of migraine with aura”. The diagnostic criteria are: (a) A seizure fulfilling the diagnostic criteria for one type of epileptic attack, and criterion B; (b) Occurring in a patient with migraine with aura, and during or within one hour after an attack of migraine with aura; (c) Not better accounted for by another ICHD-3 diagnosis [80]. Some researchers believe that the “migralepsy” sequence does not exist in clinical practice and have hypothesized that the initial part of the “migralepsy” may be an “ictal headache” followed by other ictal autonomic, sensory, motor, or psychic signs/symptoms [72].

Migraine aura is predominantly associated with visual phenomenology. Some proposed mechanisms for the pathophysiology of migraine aura are cortical depression caused by cellular potassium and glutamate efflux. Vascular changes and calcitonin gene-related peptide (CGRP) probably also contribute to migraine symptoms [81]. The similarities between headache and epilepsy also arise from genetic mutations, such as CACNA1A and ATP1A2, which are found in both disorders, as well as shared pathophysiologic mechanisms and disbalance of excitatory and inhibitory systems that result in autonomic symptoms and brain disorder. Such similarities translate into overlapping clinical manifestations and responsiveness of both conditions to antiepileptic drugs (AEDs) [73].

In 1784, John Fothergill contributed significantly by providing a detailed description of his migraine aura: “(…) It begins with a singular kind of glimmering in the sight; objects swiftly change their apparent position, surrounded with luminous angles, like that of a fortification. Giddiness comes on, head-ach and sickness [82]”.

Although classic, the “fortification” description is not always present. In fact, flashing lights are the most frequently reported visual alteration in migraine auras, though tunnel vision, dots, and non-visual auras, including sensory, motor, brainstem symptoms, and speech disturbances may also occur [73]. In occipital seizures, visual alterations are usually stereotyped, recurrent, and tend not to last more than seconds or minutes. The visual hallucinations in epilepsy are more commonly limited to small, bright spots or shapes. Complementary exams such as EEG and MRI can help in diagnosis [73]. Somatosensory auras have also been reported with migraine. The symptoms, as with epileptic auras, generally spread across parts of the body surface, with tingling (“pins and needles”), numbness, or a combination of the two [83,84]. 

In this way, it is often extremely challenging to distinguish migraines from ictal epileptic headaches in clinical practice. Painful auras of epileptic seizures should be further discussed in the literature. The negative impact of a lack of evidence-based knowledge about this theme on patients’ health should be prevented; research exploring the intricacies that underlie an ictal origin of the pain is necessary to pick up patients’ signs and symptoms early on, avoiding futile investigations and procedures that only delay progress. Accordingly, clinicians should request an EEG in patients presenting with recurrent episodes of pain, particularly if associated with neurological symptoms such as a disturbed state of consciousness. In these cases, the response to antiepileptic drugs should be closely monitored because they can provide clues leading to a specific diagnosis of the patient’s reported pain. Nevertheless, clinicians must consider the possibility of pain being the sole, or an adjunct, manifestation of epilepsy [73].

### 6.3. Thermal Auras

Thermal auras are usually described as unpleasant coldness or warmth sensations that may spread from the initial site to involve the arm, leg, ipsilateral face, or rarely bilateral parts. Mauguière et al. reported thermal auras to occur in about 11% of the patients with somatosensory auras [20]. Interestingly, thermal perceptions were always associated with either elemental paresthesias or pain. 

Thermal changes have been reported to occur in 5% of patients with SSAs and be associated with PLE [31]. ECoG studies revealed that most thermal sensations occur due to perisylvian area stimulation. Additionally, they do not provide reliable lateralization value [10,23,24]. In a previous localization study, the SII area showed the highest rate of temperature responses, with a majority of cold sensations. Anatomically, the stimulation sites eliciting temperature sensations in SII were caudal and lateral to those producing cutaneous paresthesia [47].

### 6.4. Other Somatosensory Perceptions

Somatosensory illusions can occur in about one in every ten individuals with somatosensory epilepsy in the form of a sensation of movement, swelling, or shrinkage of a body part [20,29]. A study on patients with focal epilepsies has revealed that around 9% of patients had reported the sensation of movement of the body or body part [31]. As a disturbance of body image, these illusions can be elicited by stimulating the inferior parietal lobe, postcentral gyrus, or temporal-parietal-occipital junction, mainly on the non-dominant cerebral hemisphere [51]. However, Penfield and Jasper managed to produce a sensation of part movement on stimulation anterior to the central fissure [22]. Furthermore, the isolated sensation of ocular movement has been reported in occipital lobe seizures [85].

According to Penfield and Jasper, upon stimulation of the precentral negative motor cortex, they produced a sensation of the inability to move body parts. The same sensation has been associated with discharges in the parietal lobe involving the secondary sensory and perisylvian areas [22,51]. These have been reported to occur ipsi-, contra-, and bilaterally. Thus, they do not have lateralizing value [86].

Somatognostic illusions are associated with epileptic discharges in the posterior parietal area [87]. In the study of 127 patients with SSA by Mauguire et al., 13 individuals presented with paroxysmal somatognostic dysfunction associated with paresthesia. Nine patients described a sensation of swelling in one of their hands or of their tongues. Two patients reported a sensation of shrinking of one of their upper limbs, in which one individual described a paroxysmal left hemiasomatognosia, and the other patient had brief sensations of levitation. It is worth mentioning that the authors did not find a statistical correlation between somatognostic illusions and posterior parietal lesions, which can be explained by the small sample size [20].

In a study with 82 patients with drug-resistant PLE who underwent surgical resection, 94% of the individuals presented with aura. Fifty-two patients described paresthetic SSAs, and nine patients reported distorted body images. From these, three mentioned a sensation of movement in one extremity and another a feeling that one leg was absent. A total of 80 patients underwent intraoperative cortical stimulation, and aura was reproduced in 55% of them. Interestingly, two patients reported distorted body image, one felt a twisting sensation in the contralateral extremity, and another stated that “I just swayed” following stimulation of the non-dominant inferior parietal lobule. Penfield previously described a patient that reported a “far away sensation” and “things seem distant and small” after area 5b stimulation [29,88].

Read Table 2 for a summary of the localization and lateralization of somatosensory auras.

## 7. Epilepsy Mimics and Other Considerations

### 7.1. Alice-in-Wonderland Syndrome (AIWS)

In 1955, Alice-in-Wonderland syndrome (AIWS) was defined by the psychiatrist John Todd as a neurological condition where patients experienced paroxysmal visual or somatic disturbances. Possible manifestations include sensation of floating, of a diffuse bodily disintegration, teleopsia (objects appear further away), localized disintegration, hypertrophy of certain parts of the body, telescoping, and atrophy of a certain part of the body [97]. Thus, patients with AIWS may report body-image illusions marked by distortions in mass, size, and shape. 

The name of the syndrome was inspired by the experiences of Alice, the main character in *Alice’s Adventures in Wonderland* (1865) by Lewis Carroll. It is believed that the earliest description of this syndrome was by Jean-Martin Charcot (1825–1893) during a lecture given on 22 November 1887, at the Salpêtrière. He described a 37-year-old male who suffered from macrosomatognosia, where the entire body or parts of the body were perceived as abnormally large, alongside migraine with aura. According to a study by Brigo et al., although this condition was not known then, Charcot tried to frame the varied symptoms by linking them to migraine with aura and epilepsy [96]. His diagnostic hypothesis of sensory epilepsy, nonetheless debatable, correctly suggested the presence of a cortical excitability disorder of areas responsible for the processing and perception of sensory stimuli [98,99,100]. 

In 1952, Lippman et al. noticed a temporal relationship between the body-image illusions to episodes of migraine in a case of a 38-year-old woman who reported “a sensation of the neck extending out on one side for a foot or more; at other times her hip or flank balloons out before, with, or after the headache. Very occasionally, she has an attack where she feels small—‘about one foot high’, just before with, or during the headache [99,101]”.

Matsudaira et al. reported an elderly woman with focal epilepsy presenting with AIWS. She had focal awareness seizures, turning both eyes to the left, followed by macropsia, where she saw objects, such as faces bigger on the left side. Video-EEG showed multiple focal impaired awareness seizures accompanied by sudden motion arrest and subtle body movements. Ictal EEG changes arose over the right occipital region with small periodic positive discharges with the evolution towards the right centro-parietal regions. They eventually ended with slow rhythmic waves in the right temporal region [102].

The pathophysiological explanation of AIWS is unknown. However, abnormal activation of the temporoparietal junction and temporal and occipital lobe has already been observed in neuroimaging studies. AIWS may also occur in epilepsy, encephalitis, brain lesions due to medication side effects, schizophrenia, and other psychiatric disorders. A systematic review reported that migraines (27.1%), infectious diseases (22.9%), and medication- or substance-induced (6.0%) causes are commonly associated with AIWS. Noteworthy, epilepsy is a rare cause of AIWS and was only reported in three percent of the cases [103].

### 7.2. Differential Diagnoses

Xu et al. systematically reviewed the literature on epilepsy mimics, and they found that the most common imitators of epilepsy are syncope (52.4%), psychogenic non-epileptic paroxysmal events (34.7%), and subjective non-epileptic paroxysmal symptoms (0.4%). Interestingly, a sensitivity analysis with autism, mental retardation, and learning disability revealed a strong association with subjective non-epileptic paroxysmal symptoms (2.4%), migraine (2.2%), and cerebrovascular non-epileptic paroxysmal events (0.7%) [104]. Interestingly, anoxic epileptic seizures caused by low blood flow in adults are commonly misdiagnosed with psychiatric disorders due to the biphasic sequence of syncope followed by convulsive epileptic seizure [104,105]. Table 3 provides a table with a summary of these findings.

In a prospective Austrian study about auras in sixty patients with cardiogenic and forty with non-cardiogenic syncope, the most common types of auras found were epigastric, vertiginous, visual, and somatosensory. The authors defined aura as “(…) a perceptive experience with onset immediately before losing consciousness”. A neurological examination was performed, and individuals with suspected epileptic seizures were excluded. The cardiovascular group individuals reported one-hundred and fourteen auras, with an average of 2.15 auras per patient during the study. In the noncardiac group, the total number of auras reported was one-hundred and twenty-nine, with 3.25 per patient. Vertiginous and visual auras were slightly more frequent in the cardiac group. On the other hand, somatosensory, cognitive, affective, and auditory auras were more frequent in the noncardiac group. Specifically, somatosensory auras were present in 16 (14%) individuals in the cardiac group and 23 (17.8%) individuals in the noncardiac group. The authors highlighted that although the distinction of epileptic auras from syncope-related auras is difficult, the latter is rarely associated with certain experiences commonly observed in epilepsy, such as déja vu, olfactory and gustatory sensations, scenic visual perceptions, and speech impairments. Indeed, in their study, none of the patients reported such phenomena. The study also mentioned that auras preceding syncope had poor localizing value since sensory auras, for example, are rarely limited to an individual body region. Additionally, unilateral symptoms indicate other etiologies rather than syncope [106].

In another study that aimed at outlining strategies to differentiate syncope, epileptic seizures, and psychogenic non-epileptic seizures (PNES) in patients with transient loss of consciousness, the authors mentioned the importance of clinical history, home video recordings, physical examination, and appropriate diagnostic tests to arrive at the correct diagnosis. The authors recommended that in cases of loss of consciousness due to syncope, patients may experience prodromal symptoms such as blurry vision, tunnel vision, diplopia, darkening of vision, light-headedness, nausea, diaphoresis, chest pain, and palpitations. Also, some individuals may recall a change in their posture just before the event [107]. 

The episode’s duration can help distinguish syncope from seizures since syncope is usually very brief, with a duration of approximately twenty seconds, and the recovery time is faster. However, seizures can last up to a minute, and the return to baseline or post-ictal phase could be prolonged and is usually characterized by confusion and disorientation. In PNES, the differentiation from seizure or other causes of transient loss of consciousness is not so straightforward, and more than clinical history is needed. Some aspects suggestive of PNES are speech, crying, side-to-side head movements, biting the tip of the tongue instead of biting the side of the tongue, prolonged duration of the episode, and disorganized asynchronous limb movements [107]. 

Asadi-Pooya et al. studied patients diagnosed with PNES. The authors observed that 67.1% of the patients reported having auras with their seizures. The auras were described as breathing difficulty, dizziness, headache, palpitation, vertigo, and weakness. A small percentage of the individuals described thermal sensation, nausea and abdominal discomfort, hearing voices, and visual auras. Interestingly, a logistic regression analysis revealed a statistically significant association between auras and a history of traumatic brain injury and sex [108].

Recent studies with brain MRI have also explored somatosensory network connectivity in neuropathic pain. Patients with diabetic neuropathic pain participated in a cohort study and were divided into irritable (IR, *n* = 10) and non-irritable (less preserved sensory function) (NIR, *n* = 33) nociceptor phenotypes. In situ brain MRI included 3D T1-weighted anatomical and 6 min resting-state functional MRI scans. It was observed that patients with IR have increased thalamus–insular-cortex functional connectivity. Those with the non-irritable nociceptor phenotype have greater thalamus–somatosensory-cortex functional connectivity [109].

## 8. Future Directions

Even though somatosensory auras have a high localizing value and are important in epilepsy prognostication, they remain underreported in many individuals, especially the elderly, the underserved, and those with intellectual disabilities, which could delay diagnosis and treatment. Further diagnostic methods should be developed to detect auras, especially in those populations, in association with increased awareness about those features of the disease, as they can serve as warning signs for patients with epilepsy, allowing a greater perception of self-control of the disease. In the future, large cohort studies and randomized trials with people suffering from auras should be made to develop more efficient rescue therapies for epilepsy.

## 9. Conclusions

The upper extremities are more commonly affected by somatosensory auras, followed by the lower extremities and the face. The most common type of somatosensory aura is paresthetic, followed by painful and thermal auras. In SI auras, sensations occur more commonly contralaterally, while SII auras can be ipsilateral or bilateral. Regardless of the high localizing features of somatosensory areas, cortical stimulation studies have shown overlapping sensations originating in the insula and the supplementary sensorimotor area.

## Figures and Tables

**Figure 1 medicines-10-00049-f001:**
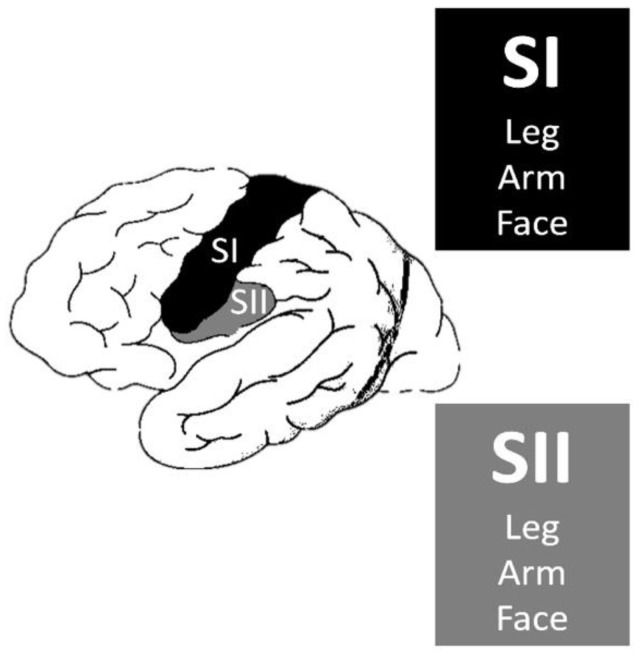
Primary somatosensory area (SI) and secondary sensory area (SII).

**Figure 2 medicines-10-00049-f002:**
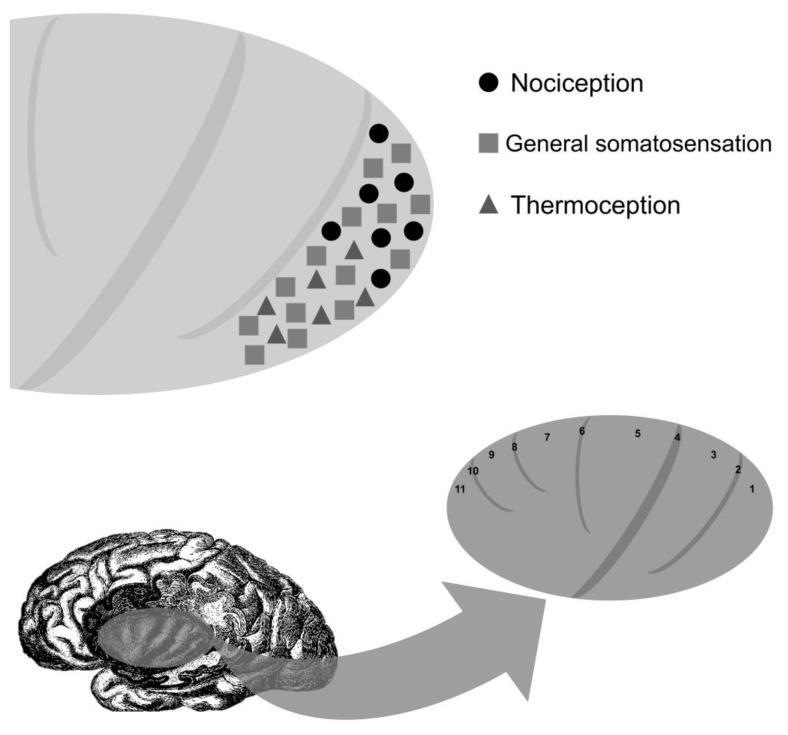
Localization of clinical responses with electrocortical stimulation of the insula. Long gyrus of the insula (1, posterior long gyrus of the insula; 2, postcentral insular sulcus; 3, anterior long gyrus of the insula), central insular sulcus (4), short gyri of the insula (5, posterior short gyrus of the insula; 6, precentral insular sulcus; 7, middle short gyrus of the insula; 8, short insular sulcus; 9, anterior short gyrus of the insula), and accessory gyrus of the insula (11) and sulcus (10).

**Figure 3 medicines-10-00049-f003:**
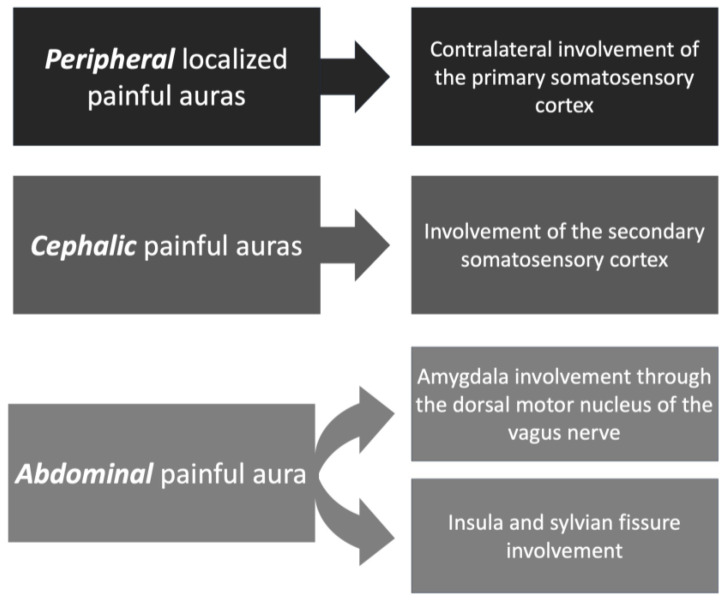
Types of painful auras and corresponding theories.

**Table 1 medicines-10-00049-t001:** Incidence of somatosensory auras throughout studies in the literature.

Reference	Incidence ^a^	Total ^b^	SSA Cases ^c^	Comment
Palmini et al. [26]	17.88%	179	32	Prospective and retrospective groups
Perven et al. [27]	7.81%	333	26	TLE surgery
Kim et al. [28]	32.50%	40	13	PLE
Salanova et al. [29]	63.41%	82	52	PLE
Afif et al. [30]	44%	25	11	DRE
Tuxhorn et al. [31]	12.50%	600	75	DRE; Seventy-seven percent reported paresthesia
Lennox et al. [10]	7.33%	750	55	Paresthesia (32) and pain (23)
Mauguiere et al. [20]	1.42%	8938	127	Ninety percent were unilateral paresthetic seizures
Cascino et al. [32]	80%	10	8	Intractable PLE
Currie et al. [33]	2.10%	666	14	TLE
Williamson et al. [34]	36.36%	11	4	PLE
Ludwig et al. [35]	31.93%	238	76	Fronto-central (20), centro-parietal (21), central (15), frontal (8), fronto-centro-temporal (8), occipital temporal (3), and with occipital bilateral synchronous abnormality (1)
Liu et al. [36]	19.53%	297	58	There were 58 individuals with SSAs, 63% presented with parietal, 27% central, 22% frontal, 18% temporal, and 0% occipital abnormalities in the electroencephalogram
Rasmussen et al. [37]	17.5%	40	7	Frontal lobe epilepsy
Quesney et al. [38]	15%	40	6	Sixty percent of the seizures arising from the parasagittal region were associated with SSAs
Yin et al. [39]	3.67%	327	12	The SSAs were most commonly related to contralateral motor areas
Balestrini et al. [40]	56.4%	172	97	Focal DRE
Janszky et al. [41]	22.22%	27	6	Resective epilepsy surgery of the frontal lobe
Chassagnon et al. [42]	19.23%	52	10	Frontal lobe DRE
Silveira et al. [43]	6%	100	6	One individual >55 years old and five between 18–45 years old

Abbreviations: DRE, drug-resistant epilepsy; PLE, parietal lobe epilepsy; SSA, somatosensory aura; TLE, temporal lobe epilepsy. ^a^ Incidence = total/SSA cases. ^b^ Total represents the number of individuals investigated in the study. ^c^ SSA cases represent the number of individuals with epilepsy who had SSA.

**Table 2 medicines-10-00049-t002:** Localization and lateralization of somatosensory auras by Rona et al. [89] and Perven et al. [90], adapted by Caprara et al.

Somatosensory Aura Type			Symptomatic Zone (Electrical Stimulation)	Epileptogenic Zone (Lobe)	Lateralization	Reference
Paresthesia	Limbs		SSI, SSII, SSMA, insula, temporal	All lobes, more often parietal or temporal	SSI: contralateral SSII, SSMA, insula: ipsilateral	Ostrowsky et al. [91]
	Head, trunk, genitals		SSI, SSII, SSMA, insula, temporal	All lobes, more often parietal or temporal	No	Kim et al. [28]
	Localized Jacksonian march		SSI, SSII, SSMA, insula, temporal	All lobes, more often parietal or temporal	No	Ajmone-Marsan et al. [21]
Pain sensation			SSI, SSII, insula	Parietal, temporal	SSI: contralateral SSII and insula: ipsilateral	Shibasaki et al. [92]
Thermal sensations		Warm	SSII, insula	Parietal	No	Bowsher et al. [93]
		Cold	SSII, insula, amygdala, anterior cingulate/mesial frontal	Temporal	No	Loddenkemper et al. [94]
Somatosensory illusions			SSI, inferior parietal lobule, temporo-parieto-occipital junction	Parietal, temporal	More frequent in the non-dominant hemisphere	Kahane et al. [95]
Sensation of movement			SSI, SSMA, primary motor cortex	Frontal, parietal	No	Romaiguere et al. [96]
Sensation of movement of the eye			Occipital lobe	Occipital	No	Holtzman et al. [84]

Abbreviations: SSI, primary somatosensory area; SSII, secondary somatosensory area; SSMA, supplementary sensorimotor area.

**Table 3 medicines-10-00049-t003:** Epilepsy Mimics in a total of 906 patients in 18 studies according to Xu et. al 2016 [104].

Category	Number of Cases	Percentage	Number of Studies
Psychogenic non-epileptic paroxysmal events	314	34.7%	12
Syncope	475	52.4%	15
Migraine	20	2.2%	4
Cerebrovascular non-epileptic paroxysmal events	6	0.7%	12
Autism, mental retardation, learning disability	29	3.2%	1
Subjective non-epileptic paroxysmal symptoms	4	0.4%	1

## Data Availability

Not applicable.

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
