# Peer review of "Somatosensory Auras in Epilepsy: A Narrative Review of the Literature"

_medicines, 2023, doi:10.3390/medicines10080049_

Round 1
Reviewer 1 Report
Review of Manuscript “Somatosensory Auras in Epilepsy: A Narrative Review of the Literature” submitted by Ana Leticia Fornari Caprara et al.
This is an interesting review which comprehensively describes somatosensory auras observed in epilepsy. The review provides detailed information, starting with the history of auras to their nowadays definition based on ILAE. The manuscript describes the most common types and subtypes of somatosensory auras and their prevalence among patients. Moreover, the authors outline the problems which can occur when auras are underestimated by patients or physicians. Information on the localization value of somatosensory auras and its consequences has been given as well. Overall, the manuscript is written in good English, and it is easy to read.
My only recommendation is for patients' numbers and percentages to be given only with numbers (not mixed words and numbers), as it is quite confusing. It should be all the same throughout the whole text.
Author Response
Review of Manuscript “Somatosensory Auras in Epilepsy: A Narrative Review of the Literature” submitted by Ana Leticia Fornari Caprara et al.
This is an interesting review which comprehensively describes somatosensory auras observed in epilepsy. The review provides detailed information, starting with the history of auras to their nowadays definition based on ILAE. The manuscript describes the most common types and subtypes of somatosensory auras and their prevalence among patients. Moreover, the authors outline the problems which can occur when auras are underestimated by patients or physicians. Information on the localization value of somatosensory auras and its consequences has been given as well. Overall, the manuscript is written in good English, and it is easy to read.
My only recommendation is for patients' numbers and percentages to be given only with numbers (not mixed words and numbers), as it is quite confusing. It should be all the same throughout the whole text.
Reply: The authors would like to thank the reviewer for the constructive comment and we addressed this question by making the recommended modifications in the text.

Reviewer 2 Report
The narrative review of the somatosensory auras in the research paper titled “Somatosensory Auras in Epilepsy: A Narrative Review of the Literature” is comprehensive and rich in scientific material. However, correction of the following is suggested. It is necessary to provide the relevance and the connect of describing the story of the Greek mythology with regards to the meaning of aura in seizure in the first paragraph in introduction. Kindly check for correctness in the highlighted sentences in the attached document. In figure 5, the authors are requested to provide the size of the circle proportionate to the percentage.

The article is well written. Kindly check the highlighted sentence in the attached pdf document for correctness.
Reviewer 3 Report
Aura is a polymorphic symptom of a number of diseases, such as epilepsy, migraine, fainting. Despite the fact that in 2017 ILAE revised the term "aura" and proposed a new interpretation of this condition as "focal awareness seizure", or "focal awareness seizure", clinicians around the world continue to actively use the term "aura" in their practice. Thus, from the point of view of practical medicine, ideas about the possible types of aura are very important for the purposes of differential diagnosis of the above diseases. In this regard, the topic of this review becomes relevant. The review manuscript is written in detail, interestingly, the authors have done a great job, analyzed the literature sources of 1880-2023, in general, the impression of the reviewed manuscript is good.
However, during the test, some observations arose:
1) The abstract is not structured. It is recommended to structure according to the following plan: background, purpose, materials and methods, results, conclusion.
2) The section "Materials and Methods" needs to be expanded, to add information about how many sources of literature were found initially and how many of these sources were included in this review.
3) Section (7.2) is devoted to the differential diagnosis of various types of aura in various diseases (epilepsy, non-epileptic seizures, fainting, migraine), the material is too large and difficult to understand, it is recommended to arrange this section in the form of a table or figure for a more visual perception of information. It makes sense to combine sections 7.1 and 7.2.
